# Crossroads of Iron Metabolism and Inflammation in Colorectal Carcinogenesis: Molecular Mechanisms and Therapeutic Perspectives

**DOI:** 10.3390/genes16101166

**Published:** 2025-10-01

**Authors:** Nahid Ahmadi, Gihani Vidanapathirana, Vinod Gopalan

**Affiliations:** 1Department of Immunology, School of Medicine, Lorestan University of Medical Sciences, Kamalvand Campus, Khorramabad 6813833946, Iran; nahidahmadiimenii@gmail.com; 2School of Medicine & Dentistry, Griffith University, Gold Coast Campus, Southport, QLD 4222, Australia; gihani.vidanapathirana@griffithuni.edu.au; 3Department of Medical Laboratory Science, Faculty of Allied Health Sciences, University of Peradeniya, Peradeniya 20400, Sri Lanka

**Keywords:** iron homeostasis, CRC, inflammatory pathways, tumour microenvironment

## Abstract

Background/Objectives: Colorectal cancer (CRC) is a leading cause of cancer-related mortality worldwide. Iron metabolism and chronic inflammation are two interrelated processes that significantly influence the initiation and progression of CRC. Iron is essential for cell proliferation, but its excess promotes oxidative stress and DNA damage, while inflammation driven by cytokine-regulated pathways accelerates tumourigenesis. We therefore conducted this narrative review to collate the available evidence on the link between iron homeostasis and inflammatory signalling in CRC and highlight potential diagnostic and therapeutic applications. Methods: This narrative review of preclinical and clinical studies explores the molecular and cellular pathways that connect iron regulation and inflammation to CRC. Key regulatory molecules, such as the transferrin receptor (TFRC), ferroportin (SLC40A1), ferritin (FTH/FTL), hepcidin, and IL-6, were reviewed. Additionally, we summarised the findings of transcriptomic, epigenomic, and proteomic studies. Relevant therapeutic approaches, including iron chelation, ferroptosis induction, and anti-inflammatory strategies, were also discussed. Results: Evidence suggests that CRC cells exhibit altered iron metabolism, marked by the upregulation of transferrin receptor (TFRC), downregulation of ferroportin, and dysregulated expression of ferritin. Inflammatory mediators such as IL-6 activate hepcidin and STAT3 signalling, which reinforce intracellular iron retention and oxidative stress. Increased immune evasion, epithelial proliferation, and genomic instability appear to be linked to the interaction between inflammation and iron metabolism. Other promising biomarkers include ferritin, hepcidin, and composite gene expression signatures; however, their clinical application remains limited. Although several preclinical studies support the use of targeted iron therapies and combination approaches with anti-inflammatory agents or immunotherapy, there is a lack of comprehensive clinical validation confirming their efficacy and safety in humans. Conclusion: Although preclinical studies suggest that iron metabolism and inflammatory signalling form an interconnected axis closely linked to CRC, translating this pathway into reliable clinical biomarkers and effective therapeutic strategies remains a significant challenge. Future biomarker-guided clinical trials are essential to determine the clinical relevance and to establish precision medicine strategies targeting the iron–inflammation crosstalk in CRC.

## 1. Introduction

Colorectal cancer (CRC) is the third most prevalent cancer and the second leading cause of cancer-related deaths worldwide. It caused about 1.9 million new cases and over 900,000 deaths in 2020 [1]. This burden is especially high in developed countries, where it is closely linked to Western diets, obesity, lack of physical activity, and an ageing population. In addition to the rising number of cases, it is challenging to identify CRC early because it consists of a diverse range of molecules and interacts with the host in complex ways [2]. There have been major advances in understanding the genetic makeup of CRC, including the identification of key mutations in the APC, KRAS, and P53 genes. Other factors, including genetics, lifestyle factors such as dietary patterns, alcohol consumption, smoking, and host metabolic and inflammatory responses, are all implicated in the initiation and maintenance of neoplastic changes in the colonic epithelium, with iron metabolism being one of the least studied factors. Although this factor is essential for maintaining cellular metabolism, its levels are regulated by iron regulatory proteins to prevent cytotoxicity.

The balance between iron surplus and deficiency plays a crucial role in tumourigenesis, progression, metastasis, and treatment response. In CRC, abnormal iron metabolism is well established, as these cells exhibit a higher demand for iron due to the rapid DNA synthesis during cell growth. Tumour microenvironment (TME) has a high demand for iron, leading to a shift toward pro-oncogenic iron loading, inflammation, and immune dysfunction [3].

Recent ground-breaking research has shown strong evidence connecting iron metabolism to the development of CRC. The adenoma–carcinoma sequence is driven by the successive accumulation of oncogenic and tumour suppressor mutations, as established by Fearon and Vogelstein’s stepwise genetic model of CRC development [4]. Advances in iron biology have established hepcidin as a key regulator of iron homeostasis, primarily by inducing the internalisation and degradation of ferroportin, which in turn controls systemic and cellular iron export [5]. Subsequently, De Domenico et al. demonstrated that the surface stability of ferroportin depends on ferroxidase activity provided by multicopper oxidases, such as ceruloplasmin. When this activity is impaired, intracellular iron retention is accelerated [6]. The concept of ferroptosis, a nonapoptotic cell death process defined by lipid peroxidation and iron dependence, was proposed by Dixon et al. [7]. Taken together, these findings provide additional evidence that conventional genetic abnormalities, dysregulated iron homeostasis, and iron-driven cell death pathways may all contribute to the initiation and progression of CRC.

Due to its multifaceted role, iron metabolism is gaining increasing significance in the progression of CRC. Owing to its essential roles in DNA synthesis and cellular oxygen transport, iron can have both beneficial and detrimental effects on genomic stability and redox homeostasis. As a potential catalyst for the Fenton reaction, iron may have a function in one-electron redox cycling. The Fenton reaction is a well-known source of hydroxyl radicals and other reactive oxygen species (ROS). This particular free radical is responsible for the oxidative damage to DNA, lipids, and proteins. This pathway is known to damage epithelial cells harbouring APC mutations through the production of two cytotoxic aldehydes, malondialdehyde (MDA) and 4-hydroxynonenal (4-HNE). When ferroportin (FPN/SLC40A1) levels are low and transferrin receptor 1 (TfR1) levels are high, iron accumulates in tumour-associated macrophages (TAMs) and colon epithelial cells (TECs), preventing iron from being exported normally. The iron stays inside the cells in this manner.

This iron retention not only supports tumour cell growth and survival but also alters the immune response by shifting TAMs toward a pro-tumourigenic M2-like phenotype and suppressing the activity of cytotoxic T cells [8]. Additionally, CRC cells often resist iron-dependent ferroptosis by continually activating antioxidant defence systems, such as glutathione peroxidase 4 (GPX4) and nuclear factor erythroid 2-related factor 2 (NRF2). As a result, tumour cells are better able to withstand oxidative damage. CRC iron metabolism is, therefore, a complex system that promotes cancer via immune control, metabolic reprogramming, and redox biology [9].

## 2. Iron Metabolism in the Colon and CRC

As a complex and highly regulated system, iron homeostasis protects individuals against the harmful effects of excessive iron and ensures that their metabolic processes function properly. The duodenum plays a crucial role in absorbing iron from the diet. Interconnected systems, including the digestive tract, liver, spleen, bone marrow, and systemic immune mediators, regulate iron absorption and utilisation. In normal physiology, enterocytes at the brush border of the duodenum absorb dietary iron either as heme iron or in the ferrous form (Fe^2+^) [10]. Divalent metal transporter 1 (DMT1) is a protein present on the apical membrane of the enterocytes, responsible for the transportation of Fe^2+^ across cell membranes. Iron can either be stored as ferritin (a protein complex made of heavy and light subunits) in the labile iron pool or it can be exported into the bloodstream through FPN/SLC40A1, the only known iron exporter. Hephaestin oxidises iron that has been exported to Fe^3+^ and binds it to transferrin. This process is tightly regulated at multiple levels, including gene expression, RNA processing, and protein modification. Hypoxia, inflammation, stress from cancer, and cellular iron levels are some of the other factors that might affect it [11].

### 2.1. Physiological Pathways and Regulatory Mechanisms

In CRC, the iron homeostasis is disturbed, supporting the malignant transformation of colon epithelial cells. Tumour cells have a higher demand for iron to support fast growth, DNA synthesis, and mitochondrial activity. This leads to changes in their iron metabolism. As a result, CRC epithelial cells have much higher levels of transferrin receptor 1 compared to adjacent normal tissues [12]. Estêvão et al. [13] found that transferrin receptor 1 (TFRC) expression was transcriptionally induced in CRC tissues and linked to the consensus molecular subtype CMS2, which is characterised by active Wnt signalling and rapid tumour cell growth. Under low oxygen conditions, HIF-1α directly initiates the transcription of TFRC, which binds to hypoxia-responsive elements in the TFRC promoter. This facilitates continued iron absorption by the body, even under conditions of low oxygen or nutrient availability. Hypoxia triggers a temporary adaptive response to restore oxygen levels to normal in healthy physiological conditions. In contrast, within the TME of colorectal cancer, hypoxia is consistently present and sustained. As a result, TFRC is activated, facilitating tumour cells to absorb iron bound to transferrin. The increase in intracellular labile iron not only fulfils metabolic demands but also speeds up cell growth by supporting DNA synthesis and mitochondrial respiration. Too much labile iron also causes the Fenton reaction to make ROS, which can damage DNA and make the genome unstable. Ferritin is the major iron-storage protein, composed of heavy (FTH1) and light (FTL) chains, and can securely sequester up to 4500 iron atoms in a non-reactive, bioavailable form. Ferritin is often overexpressed in CRC. Ferritin not only protects against iron overload but also supports tumour growth, survival, and immune evasion by regulating iron availability and modulating the inflammatory environment within the TME [14]. Several CRC transcriptomic datasets have shown that FTH1 is expressed at higher levels, and this upregulation has been confirmed at the protein level through immunohistochemistry [15].

Nrf2, NF-κB, and HIF-1α regulate ferritin transcription, while iron regulatory protein–iron-responsive element (IRP–IRE) interactions control its post-transcriptional regulation. Notably, HIF-2α mRNA contains an IRE sequence that allows direct regulation by IRPs [16,17,18]. In tumour cells, the normal IRP-mediated repression of ferritin under low iron conditions is disrupted, allowing ferritin to be produced even when iron is scarce. This is likely due to oncogenic signalling and exposure to inflammatory cytokines. High ferritin expression in CRC correlates with enhanced tumour invasiveness, reduced sensitivity to ferroptosis-mediated cell death, and unfavourable clinical outcomes in patients. Ferritin may also change the immune system by preventing dendritic cells from maturing and T-cells from activating, which helps the tumour cells to avoid the immune system [19] (Figure 1).

Iron regulatory proteins (IRP1 and IRP2) oversee the whole iron regulatory network. They regulate the stability and translation efficiency of mRNAs containing iron-responsive elements (IREs). Under conditions of iron deficiency, iron regulatory proteins (IRPs) bind to iron-responsive elements (IREs) located in the 5′ or 3′ untranslated regions (UTRs) of target mRNAs. This maintains TFRC and DMT1 transcripts and prevents the translation of ferritin and FPN. This post-transcriptional regulation is disrupted in CRC. Despite elevated intracellular iron levels, tumour tissues continue to show increased expression of IRP2, indicating that the normal cellular mechanisms for sensing and responding to iron levels are impaired. Xenograft studies have shown that tumour growth is accelerated when the level of IRP2 in CRC cells is increased. Increased IRP2 levels stabilise TFRC mRNA, leading to sustained expression of transferrin receptors and, consequently, enhanced iron uptake by tumour cells. These findings highlight the significance of IRP dysregulation in promoting tumour metabolism and demonstrate how aberrant IRP2 activity increases iron dependence in CRC cells [20,21,22,23].

Hypoxia-inducible factor for transcription 1-alpha (HIF-1α) links low oxygen stress in CRC with iron metabolism. When the oxygen level is insufficient, HIF-1α stabilises and activates many target genes that aid in iron acquisition (such as TFRC and DMT1), sugar breakdown, and the formation of new blood vessels. Fagundes et al. [24] has shown that iron-related gene expression in CRC cells is regulated by HIF-1α, particularly in CMS3 tumours that have metabolic abnormalities. Evidence for direct HIF-1α binding to the TFRC or FTH1 promoters has primarily been demonstrated in intestinal or inflammation-related models, rather than CRC-specific ChIP-seq studies. In CRC, TFRC upregulation is frequently driven by oncogenic pathways (e.g., Wnt/β-catenin) and may be modulated by hypoxia–inflammation contexts. To rapidly adapt to the hypoxic TME, cells activate a transcriptional programme that limits iron efflux, increases iron uptake, and preserves iron-dependent metabolic processes. CRC disrupts the normal iron homeostasis in the body. This is partly due to HIF-1α being present in high levels, which increases TFRC expression and enables it to absorb more iron to meet its demands for rapid growth and metabolism. Increased labile iron leads to more ROS and damages DNA. Conversely, excessive ferritin not only sequesters iron to prevent toxicity but also promotes tumour growth, facilitates immune evasion, and contributes to resistance against ferroptosis. Even in the cells with elevated iron levels, TFRC translation continues due to aberrant IRP2 activity. As a result, altered iron regulation is crucial to the development of CRC.

Redox reactions involving iron and other metals can generate ROS, which are toxic to living organisms. Iron acts as a carcinogenic cofactor by amplifying oxidative stress through the Fenton reaction, which generates ROS that initiate and sustain DNA damage, lipid peroxidation, and epigenetic reprogramming. The hydroxyl radical (OH), which originates from Fenton chemistry, involving Fe^2+^ (Fe^2+^ + H_2_O_2_ → Fe^3+^ + •OH + OH^−^), can break DNA strands and oxidise important macromolecules in cells [25] (Figure 2).

This process is pathologically amplified in colorectal tissues, particularly under conditions of high dietary heme intake. Ijssennagger et al. [26] conducted a foundational study demonstrating that mice fed a heme-rich diet exhibited elevated ROS levels in the colon and increased mucosal cytotoxicity. These changes were associated with crypt hyperproliferation and a marked increase in aberrant crypt foci (ACF), recognised as precancerous lesions in CRC. The study further confirmed that heme iron induced lipid peroxidation within the colonic mucosa. This was demonstrated by higher levels of thiobarbituric acid-reactive substances (TBARS) and 4-HNE, which are known to form DNA adducts and mutagenic cross-links. These results show a direct link between dietary heme, the production of ROS, and early molecular lesions that are linked to CRC.

### 2.2. Role of Ferroportin, Hepcidin, Ferritin, and Other Iron-Metabolism-Related Proteins

Studies demonstrating pathological iron accumulation within tumour tissues provide additional evidence supporting iron’s contributory role in promoting tumourigenesis. De Domenico et al. [27] found that FPN expression was significantly lower in CRC tissues. This suggests that hepcidin-driven degradation of ferroportin is a key mechanism preventing iron efflux from neoplastic cells. Accumulation of iron in the cell has biological effects that exacerbate oxidative DNA damage, primarily through ROS-driven lesions such as 8-oxo-2′-deoxyguanosine (8-oxo-dG), abasic sites, and DNA strand breaks. Failure to accurately repair these DNA lesions can result in mutagenesis and chromosomal instability, both of which are key drivers of colorectal carcinogenesis [28,29,30,31,32]. Chua et al. [33] demonstrated in vitro that supplementing the media with iron, in which CRC cells were grown, caused DNA double-strand breaks (as shown by γ-H2AX foci) and micronucleus formation in a dose-dependent way. These two observations indicate that the genome is unstable. The Hfe^−^/^−^ mice model of hereditary hemochromatosis has been shown to enhance the risk of CRC associated with colitis. They showed increased tumour burden and higher levels of oxidative stress markers such as Nrf2 and HO-1 compared to controls with wild-type genotypes. Iron overload leads to immune system problems and oxidative stress, which, if persistent, can encourage tumour formation. Colonic epithelial cells are vulnerable to lipid oxidation by ROS due to the high content of polyunsaturated fatty acids (PUFAs). This mechanism yields bioactive aldehydes such as 4-HNE and MDA. These chemicals disrupt critical cellular signalling pathways and create mutagenic adducts when they interact with DNA bases and proteins. Bastide et al. [34] found that heme-fed mice had elevated levels of amphiregulin and MDA adducts. Amphiregulin is a growth factor which may support the survival of initiating cells and promote the proliferation of epithelial cells. This supports the hypothesis that lipid oxidation products could serve as non-invasive biomarkers for early cancer progression [35].

Epigenetic landscapes are also significantly affected by iron excess, which changes DNA methylation and histone modification in CRC. Iron is a cofactor for the Jumonji-C domain histone demethylases and the ten-eleven translocation (TET) family of DNA demethylases. Excessive iron can overactivate these enzymes, leading to DNA hypomethylation and increased gene expression. Treatment of CRC cells with iron may reduce overall levels of 5-methylcytosine (5mC) and demethylates the promoters of pro-inflammatory genes such as IL6 and COX2. This sustained activation of the inflammatory axis [36,37,38]. At the same time, histone analysis revealed that the repressive H3K27me3 mark was reduced, allowing cell cycle regulators such as CDK1 and Cyclin D1 to be transcribed abnormally. When treated with the iron chelator deferoxamine (DFO), these changes could be reversed. This suggests that iron is a contributing factor to epigenetic dysregulation. Supporting studies have shown that high-iron CRC tumours exhibit an epigenetic signature characterised by chromatin relaxation and reactivation of enhancers associated with iron metabolism and immune response genes [39].

Iron also affects the regulatory networks of non-coding RNAs, especially microRNAs (miRNAs). Subsequently, these microRNAs influence both the behaviour and progression of malignancies. An oncogenic microRNA, known as miR-21, affects the action of tumour suppressor genes PTEN, PDCD4, and RECK. Research has shown that iron exposure in CRC increases levels of this miRNA. Organoids derived from CRC patients show how different tumors react to the same therapy. Also, there’s mounting evidence that too much iron—specifically dietary heme iron—may alter colonic epithelial gene activity, activate miR-21 and other oncogenic microRNAs, and promote lipid peroxidation, which in turn aid in tumor development. There has been an association between 5-fluorouracil (5-FU), a foundational drug in colorectal cancer treatment, and miR-21 resistance. Taken together, this results raise the intriguing prospect that iron overload may promote tumor formation and treatment resistance by doing more than just damaging lipids and DNA via oxidative stress; it may also permanently change gene expression [34,40,41,42,43,44].

CRC iron metabolism regulators include transporters (TFRC, FPN), storage proteins (FTH1/FTL), post-transcriptional regulators (IRP1/2), and transcriptional factors (HIF-1α), as illustrated in Table 1. Under normal conditions, these molecules maintain iron homeostasis; however, in CRC, their regulation is disrupted. This imbalance leads to increased iron levels, greater susceptibility to oxidative stress, and accelerated tumour development.

## 3. Chronic Inflammation and CRC

The chronic inflammation that afflicts inflammatory bowel disease (IBD) patients, especially those with Crohn’s disease (CD) or ulcerative colitis (UC), is recognised to elevate the risk of CRC [54]. There is both epidemiological and mechanistic evidence linking long-term inflammation of the mucosa to colorectal neoplasia. For example, patients with UC have a higher risk of CRC over time, with a cumulative incidence of almost 18% after three decades of disease progression, especially in those with pancolitis and primary sclerosing cholangitis (PSC) simultaneously [55]. From a histopathological perspective, colitis-associated colorectal cancer (CA-CRC) differs from sporadic CRC, originating in flat, diffusely dysplastic mucosa, rather than arising from an adenomatous precursor lesion. The tumourigenic microenvironment in CA-CRC is characterised by mutations, aberrant regeneration, constant activation of innate and adaptive immunological pathways, proliferation of pro-inflammatory cytokines, and reactive oxygen and nitrogen species (ROS/RNS) that harm epithelial cells [56].

Choi et al. [57] explained that the inflammatory environment in IBD leads to repeated cycles of injury and regeneration that favour the survival of epithelial clones with DNA repair defects, a higher mutational burden, and a survival advantage. Li et al. [58] found that CA-CRC and spontaneous CRC vary significantly in terms of immune cell types and molecular markers. The role of macrophages and neutrophils in establishing an immunosuppressive and genotoxic microenvironment has become increasingly evident. Chronic inflammatory stress causes epithelial cells to adopt mesenchymal cell traits, making them resistant to cell death. These changes are linked to aggressive tumour behaviour and the spread of cancer.

### Overview of Key Inflammatory Mediators (IL-6, TNF-α, COX-2)

The multi-functional cytokine interleukin-6 (IL-6) regulates immune system function, promotes blood cell production, and promotes tumour growth. Phosphorylation of Signal Transducer and Activator of Transcription 3 (STAT3) and activation of Janus kinases (JAKs) are two mechanisms by which interleukin-6 (IL-6) promotes tumour development in cancer. Upon activation, STAT3 translocates to the nucleus, where it promotes the expression of genes involved in cell proliferation (e.g., cyclin D1), survival (e.g., Bcl-XL), angiogenesis (e.g., VEGF), and immune evasion. In colitis-associated cancer, IL-6 signalling is predominantly activated in intestinal epithelium and tumour-associated immune cells. Ibrahim et al. [59] demonstrated that IL-6 levels are elevated in colitic mucosa, which is associated with increased STAT3 activation and cancer growth. Researchers found that STAT3 was consistently active in epithelial cells and infiltrating myeloid populations, particularly in macrophages and dendritic cells. This suggests a systemic amplification loop.

Another gp130 ligand secreted by CAFs, IL-11, amplifies the pro-tumourigenic effects of IL-6/STAT3 signalling. In colonic epithelial cells, this ligand triggers a strong anti-apoptotic and mitogenic response. In colitis-associated tumorigenesis and human CRC cohorts, heightened IL-11 signaling tracks with worse outcomes and promotes tumor cell survival by engaging STAT3—a pathway that drives target genes relevant to CRC biology, including SOCS3 and surviving. Notably, inflammation also leaves an epigenetic imprint on intestinal stem cells (ISCs): recent work shows persistent, inflammation-induced remodeling of ISC chromatin/DNA methylation, and complementary studies indicate that IL-6 can modulate DNA methylation programs in colon epithelium. These observations support the idea that cytokine-driven IL-6/IL-11 → STAT3 signaling interfaces with DNA methylation machinery in stem-like intestinal cells, potentially contributing to early epigenetic reprogramming in colitis-associated CRC [60,61,62,63,64,65,66].

Tumour necrosis factor-alpha (TNF-α) is a common cytokine that causes both acute and chronic inflammation by activating the NF-κB transcription factor complex through the TNFR1/TRADD/TRAF2/IKK signalling axis. When NF-κB is activated in the intestinal epithelium and lamina propria cells, it induces the transcription of a wide range of genes that control inflammation (IL-1β, IL-6), prevent cell death (Bcl-2, cIAP1), and regulate the cell cycle (Cyclin D1). Waldner et al. [67] showed that dysplastic cells rely on sustained NF-κB activity within colonic epithelial cells to survive under conditions of inflammatory stress. In mouse models of colitis-induced CRC, removing IKKβ from intestinal epithelial cells also resulted in a considerable decrease in tumour burden. This indicates that epithelial NF-κB is essential for initiating and sustaining tumour growth.

NF-κB controls the expression of COX-2, inducible nitric oxide synthase (iNOS), and matrix metalloproteinases (MMPs) in immune cells. Of these proteins, some help with tissue remodelling and angiogenesis. The study by Zhao et al. [68] confirmed this by demonstrating that macrophages and neutrophils invading dysplastic lesions showed NF-κB-dependent upregulation of COX-2 expression. There is also critical functional crosstalk between NF-κB and STAT3, which plays a key role in sustaining inflammation and promoting tumour development. IL-6, produced downstream of NF-κB activation, stimulates STAT3, which in turn suppresses NF-κB inhibitors such as IκBα. This creates a feed-forward loop that sustains chronic inflammation and promotes tumour growth. COX-2 (encoded by PTGS2) is an enzyme that can be turned on or off. It converts arachidonic acid into prostaglandins, primarily PGE2, which bind to EP receptors on epithelial and stromal cells to stimulate growth, movement, and the formation of new blood vessels. There is little COX-2 in normal colonic mucosa, but it becomes significantly hyperactive during inflammation and cancer growth. Paul et al. [69] showed that COX-2 expression was significantly higher in IBD tissues, even before the onset of dysplasia. This expression becomes even stronger in high-grade lesions and invasive carcinomas. PGE2 activates the PI3K/AKT and β-catenin pathways through EP4, which helps transformed epithelial cells survive and proliferate.

Ma et al. [70] also revealed that COX-2 produced by macrophages contributes to the recruitment of immunosuppressive myeloid-derived suppressor cells (MDSCs), which reprogramme the immune environment to become more tolerant of tumour growth. PGE2 is important because it weakens the immune response against tumours by blocking the activity of cytotoxic T cells and making regulatory T cells (Tregs) more active in suppressing the immune system. COX-2 is a key inflammatory driver of CA-CRC, and a proven chemoprevention target due to its angiogenic and anti-apoptotic properties. For example, NSAIDs like celecoxib have been shown to reduce polyp burden (Table 2).

## 4. Role of Immune Cells and the Tumour Immune Microenvironment

Various immune cell types significantly influence TME, which is characteristically inflammatory in colitis-associated colorectal cancer (CA-CRC). These cell types include T lymphocytes, neutrophils, and macrophages. Potential regulators of macrophage polarisation include environmental variables. ROS are generated, and tissue damage occurs through the typical activation of M1 macrophages by TNF-α, IL-1α, and iNOS. Healing, immunological suppression, and fibrosis are all ameliorated by alternatively activated M2 macrophages. Substances including IL-10, VEFG, and TGF-β are released by different types of TAMs, which in turn promote tumour growth. Studies have shown that in advanced dysplastic tissues, macrophages undergo a phenotypic switch from the pro-inflammatory M1 state to the immunosuppressive M2 state. This suggests that macrophage plasticity is crucial for the initiation and growth of tumours [81].

Neutrophils are typically regarded as short-term responders; however, they remain in the IBD mucosa and contribute to cancer growth through degranulation and the formation of neutrophil extracellular traps (NETs). These structures have myeloperoxidase and proteases in them that damage epithelial DNA and break down the extracellular matrix, making it easier for cells to invade [82]. Wang et al. [83] reported an increased accumulation of neutrophils at the invasive fronts of colitis-associated CA-CRC lesions. This was associated with higher levels of MMP-9 and lower levels of epithelial barrier proteins, such as E-cadherin.

T cells, especially regulatory T cells (Tregs) and Th17 cells, play a significant role in modulating the inflammatory response. In CA-CRC, FOXP3+ Tregs suppress the immune system’s ability to fight cancer, while IL-17-producing Th17 cells promote the growth of epithelial cells, the formation of blood vessels, and cell resistance to death. The IL-23/IL-17 axis is highly expressed in the dysplastic mucosa of UC patients and plays a crucial role in creating a cytokine environment that promotes tumour growth. Researchers found that RORγt and IL-23R were more active in mucosal CD4^+^ T cells, which is consistent with the fact that IL-17-rich tissues are associated with increased tumour burden [84] (Table 3).

The coordinated yet dysregulated activity of macrophages, neutrophils, and T cells creates a pro-tumourigenic inflammatory network in CA-CRC. Macrophages are the main controllers of this network because they can change shape. They can undergo transition from an M1 phenotype, which causes oxidative DNA damage through the activation of TNF-α, IL-1β, and iNOS, to an M2 phenotype that promotes angiogenesis, fibrosis, and immune suppression through the secretion of IL-10, VEGF, and TGF-β [85]. In cases of extreme dysplasia, tumour cells undergo this phenotypic change, which allows them to persist and avoid detection by the immune system. Inflamed mucosa is an ideal environment for neutrophils to secrete myeloperoxidase and protease-rich neutrophil extracellular traps (NETs), which accelerate cancer growth [86]. This occurs despite neutrophils being known primarily as rapid responders in inflammation. In addition to damaging epithelial cell DNA, they also contribute to tumour progression by degrading the extracellular matrix, facilitating invasion and tissue remodelling. Coupling with MMP-9 overexpression, they enable tumour invasion at the leading edge, which is often accompanied by the breakdown of epithelial junctions, such as E-cadherin. When T cells alter the balance between chronic inflammation and anti-tumour immunity, they also modify this environment. By reducing the activity of cytotoxic T cells, FOXP3+ regulatory T cells prevent the immune system from targeting malignancies. The formation of blood vessels, the expansion of epithelial cells, and cell resistance to apoptosis are all attributed to Th17 cells that generate IL-17 and are activated by the IL-23/IL-17 axis. This Th17-dominant cytokine environment, characterised by high levels of IL-17, RORγt, and IL-23R expression in mucosal CD4^+^ T cells, creates a feedback loop that perpetuates chronic inflammation and facilitates the transformation of cancer cells [87]. The interactions between these immune cell subsets and their soluble mediators create a microenvironment that not only supports but also accelerates the initiation, growth, and spread of CA-CRC.

**Table 3 genes-16-01166-t003:** Immune Cells in CA-CRC Tumour Microenvironment.

Immune Cell	Phenotype/Key Actions	Key Mediators/Signals	Recruitment/Localization	Markers/Subtypes	Impact in IBD → CA-CRC Progression	Therapeutic Targets/Strategies
Macrophages (TAMs)	M1 → M2 shift; diverse M2b/M2c/M2d subtypes → pro-angiogenic, immunosuppressive effects	TNF-α, IL-1β, iNOS (M1); IL-10, TGF-β, VEGF (M2)	Via CCL2/VEGF from inflamed mucosa; enriched at dysplasia/tumour fronts	CD163, CD206, SPP1^+^, Tie2^+^, F4/80^+^Ly6C^hi^	M1 initiates ROS/inflammation; M2 supports tumour growth, immune evasion, angiogenesis	CSF1R inhibitors, TAM re-polarisation, PD-L1/Siglec-15 targeting [88]
Neutrophils (TANs)	NET-forming pro-tumour TANs; some CD66b^+^/CD177^+^ subsets with potential protective roles	NET components (MPO, proteases), MMP-9; IL-23, CXCL1/CXCR2	At invasive fronts, recruited via CXCR2 ligands in dysplastic mucosa	CD66b^+^, CD177^+^; cit-histone H3 (NET marker)	NETs promote epithelial DNA damage and matrix breakdown → tumour invasion; some subsets may restrain tumour	CXCR2 antagonists; NET inhibitors (DNase, PAD4 inhibitors); subset modulation [89]
T cells (Tregs, Th17, Th1/CD8)	FOXP3^+^ Tregs (immunosuppressive), Th17 (IL-17^+^, pro-tumour), Th1/CD8 anti-tumour but suppressed	IL-23/IL-23R, RORγt (Th17); IL-12, IFN-γ (Th1); PD-L1 on TAMs	Treg & Th17 expand in dysplastic mucosa; CD8^+^ T cells infiltrate the tumour but are often exhausted	Tregs: FOXP3, RORγt^+^ Tregs; Th17: IL-17^+^, IL-23R^+^; Th1: T-bet^+^ IFN-γ producers; exhaustion markers: TOX, TIGIT	Tregs suppress CD8^+^ cytotoxicity; Th17 promotes growth, angiogenesis; Th1 responses blunted; exhausted T cells reduce anti-tumour immunity	Target IL-23/IL-17 axis; deplete Tregs or IL-17^+^ RORγt^+^ subsets; reinvigorate CD8^+^ T cells; checkpoint inhibitors [90]
Dendritic Cells (DCs, pDCs, cDCs)	Initiate and perpetuate pro-inflammatory carcinogenic programmes; pDCs suppress MDSC expansion	Antigen presentation via MHC II; CCL5 (on TADCs); pDC-MDSC regulatory interactions	Infiltrate inflamed colonic mucosa; tumour-associated DCs accumulate in TME	Conventional DCs (cDC), plasmacytoid (pDC), tumour-associated DC (TADC)	DCs can drive early carcinogenic inflammation; pDCs may counter tumour-promoting MDSCs; high DC infiltration is sometimes prognostic	DC-based vaccines; pDC modulation to reduce MDSCs; targeting TADC CCL5 pathways [91,92]
Natural Killer (NK) cells	Anti-tumour cytotoxicity via direct killing and ADCC; often depleted or dysfunctional in CA-CRC	Activating/inhibitory receptors (e.g., NKG2D), cytokines like IL-15	Low presence in CRC tissue; potentially recruited by IL-15 signalling	NK cells with NKG2D, CD56^+^; ADCC-capable NK subsets	NK cells contribute to tumour cell lysis but are often rare/dysfunctional in CA-CRC	Therapies enhancing NK infiltration/function (e.g., IL-15), ADCC-supporting antibodies [93]
MDSCs (Myeloid-Derived Suppressor Cells)	Immunosuppressive regulatory cells from myeloid lineage; suppress T, NK, DC responses	GM-CSF, G-CSF, IL-6, IL-10, TGF-β, ROS, NO	Expand systemically and infiltrate tumour via cytokine milieu in chronic inflammation	Subsets: monocytic M-MDSC, granulocytic G-MDSC; markers: CD38, PDL-1, LOX-1	Suppress CD8^+^ T cells, NK and DC function; promote tumour angiogenesis, metastasis, therapy resistance	Target STAT3, c/EBP, metabolic pathways; reduce MDSC numbers or inhibit suppressive mediators [94]
Regulatory B cells (Bregs)	IL-10, IL-35, TGF-β producers; can express Granzyme B; inhibit effector T and NK cells	IL-10, IL-35, TGF-β; PD-L1; CD39/CD73; Granzyme B	Infiltrate inflamed/tumour sites; interact with Tregs and T cells	B10 (CD24^hiCD27^+^), CD19^+^CD38^+^, PD-L1^+^, GrB^+^ Bregs	Promote immune suppression, enhance Treg activity, reduce NK/CD8^+^ cytotoxicity in TME	Deplete Bregs or inhibit IL-10/TGF-β; block PD-L1 on Bregs; modulate B-cell activation pathways [95]

## 5. Interplay Between Iron Metabolism and Inflammation in CRC

The two-way relationship between iron metabolism and inflammation is a crucial aspect of CRC development, particularly in cases where inflammation is the primary cause, such as CA-CRC. Iron metabolism and inflammation are closely interconnected in the context of tumour biology and should not be viewed as independent processes; they are closely linked through molecular crosstalk between cytokines, iron regulatory proteins, and immune-modulatory iron-binding molecules, such as ferritin. This crosstalk network not only promotes the proliferation and mutation of epithelial cells but also contributes to immune suppression, thereby facilitating tumour immune evasion [12].

### 5.1. How Inflammation Alters Iron Homeostasis

Interleukin-6 (IL-6) regulates hepcidin (HAMP), a crucial molecule that links inflammation and iron balance. Hepcidin is a peptide hormone made in the liver that attaches to FPN on macrophages, enterocytes, and epithelial cells. This causes the iron exporter to enter the cell and break down, resulting in less iron being released from the cell. IL-6 activates the JAK/STAT3 pathway during inflammation, leading to the production of hepcidin. This makes it harder for the body to absorb iron, which is a sign of anaemia of inflammation. In CRC, especially CA-CRC, this pathway is reprogrammed to retain iron in a small area within epithelial cells and TAMs. Wang et al. explained that elevated levels of IL-6 in inflamed colorectal mucosa are directly associated with increased hepatic and epithelial hepcidin expression, which promotes iron retention within dysplastic crypts [96] (Figure 3).

This inflammation-driven regulation of hepcidin serves not only as a host defence mechanism against infection but also, within the tumour microenvironment, contributes to the formation of a niche that promotes cellular proliferation and oxidative stress. Iron that is retained by hepcidin-mediated ferroportin degradation increases the labile iron pool, which facilitates the Fenton reaction and generates ROS, thereby perpetuating DNA damage and NF-κB activation. As demonstrated by Chua et al., dietary iron promotes colitis-associated colorectal cancer (CA-CRC) by increasing mucosal levels of IL-6 and IL-11, thereby activating STAT3 signaling and establishing a self-reinforcing inflammatory–iron loop that sustains tumorigenesis [98].

### 5.2. Anaemia of Inflammation in the CRC Context

Chronic inflammation alters the distribution of iron throughout the body and within cells by modifying the function of proteins that regulate iron metabolism. In CA-CRC, pro-inflammatory cytokines such as IL-1β and TNF-α limit the intestines’ ability to absorb iron and cause macrophages to retain iron, thereby moving it from the blood into tumour-associated stromal compartments [99]. Researchers revealed that macrophages and epithelial cells in dysplastic areas had higher levels of SLC11A1 (also known as NRAMP1) and DMT1. This suggests that these cells uptake increased amounts of iron from dietary sources as well as through phagocytic activity [100].

#### 5.2.1. Dysregulation of the Hepcidin–Ferroportin Axis in CRC-Associated Anaemia

Anaemia of chronic disease is a frequent systemic manifestation in CRC and is tightly linked to alterations in the hepcidin–ferroportin regulatory pathway. Under physiological conditions, hepcidin, a hepatic peptide hormone, binds to ferroportin—the only known iron exporter—causing its internalisation and degradation. This interaction prevents excessive iron release into circulation, maintaining systemic iron balance.

In CRC, chronic inflammation drives pathological hepcidin overexpression, primarily through IL-6–mediated activation of the JAK/STAT3 pathway [101]. Excess hepcidin decreases ferroportin expression in enterocytes, macrophages, and tumour-associated epithelial cells, leading to iron sequestration and reduced circulating iron levels. The paradox is that while tumour cells exploit retained intracellular iron for proliferation and metabolic reprogramming, patients develop systemic iron-restricted anaemia [102].

#### 5.2.2. Clinical Consequences

CRC-related ACD contributes to fatigue, reduced quality of life, and diminished therapeutic responses. Additionally, systemic anaemia exacerbates hypoxia in the tumour microenvironment, stabilising HIF-1α and further promoting angiogenesis, metabolic adaptation, and tumour aggressiveness [103]. Current therapeutic approaches include iron supplementation, erythropoiesis-stimulating agents, and anti-inflammatory strategies. Oral iron supplementation is often limited by poor absorption and gastrointestinal intolerance, whereas intravenous formulations such as ferric carboxymaltose or iron sucrose have shown greater efficacy in CRC patients [104]. Recombinant erythropoietin can be used in selected cases, but its use is constrained by risks of thrombosis and potential tumour progression [105]. Anti-inflammatory strategies, particularly those targeting upstream regulators of hepcidin such as IL-6 signalling, have demonstrated promise; for instance, tocilizumab (anti–IL-6R) has been shown to reduce hepcidin expression and restore ferroportin activity in preclinical models [106]. Emerging directions focus on direct modulation of the hepcidin–ferroportin axis, including the use of hepcidin antagonists such as neutralising antibodies and spiegelmers to release trapped iron into circulation, and ferroportin stabilisers designed to prevent hepcidin-induced degradation and maintain iron export. Combination therapies are also under exploration, integrating iron chelators like deferoxamine with anti-inflammatory agents or immune checkpoint inhibitors to simultaneously reduce tumour-promoting iron retention and correct systemic anaemia.

### 5.3. Ferritin as an Immune Modulator

The ferritin expression in CRC is context-dependent: It may be upregulated in tumour-associated macrophages and detectable in serum, yet downregulated intracellularly in specific subsets of tumour epithelial cells. These divergent patterns likely reflect heterogeneity in the tumour microenvironment and inflammatory status.

Ferritin, especially the heavy chain (FTH1), has two key functions: it stores iron and influences the immune system. Ferritin is more than just a protein that stores iron in the cytoplasm and protects cells from oxidative damage [107] (Figure 4).

Ferritin also has paracrine immunological effects. When tumour cells and activated macrophages release extracellular ferritin, it binds to receptors like Tim-2 and Scara5 on immune cells, changing their function. High levels of ferritin in CRC have been associated with a weakened immune system, particularly in CD8^+^ T cells and NK cells. In their study of IBD-associated tumours, Hu et al. [109] found that levels of FTH1 and ferritin light chain (FTL) were significantly higher in tumours infiltrated by immune cells compared to sporadic CRC. At the same time, pathways involved in T cell activation and antigen processing were suppressed, suggesting an inverse relationship between ferritin expression and anti-tumour immune responses. Additionally, ferritin from macrophages can inhibit the maturation and presentation of antigens by dendritic cells, thereby increasing the likelihood of T cells developing into regulatory and Th17 phenotypes. These features are more commonly observed in CA-CRC but are less prevalent in sporadic cases.

### 5.4. Crosstalk Contributing to Immune Evasion and Tumour Progression

Iron levels influence the immunological microenvironment as well as tumour cell metabolism. The immune system may be able to evade detection when intracellular iron levels are high, since MHC-I expression decreases and ICD becomes less effective at identifying infections. Once malignancies have invaded, ferritin levels rise, and the efficacy of cytotoxic T cell activity declines due to the presence of immunosuppressive molecules, such as PD-L1 and IDO1. In CRC, HIF-2α—whose mRNA contains an iron-responsive element (IRE)—drives iron uptake programs (e.g., DMT1) and promotes tumorigenesis; STAT3 signaling interconnects with iron pathways and regulates hepcidin (HAMP) expression, while CRC cells frequently express hepcidin, fostering iron retention and an immunosuppressive milieu. Collectively, these axes (hepcidin–STAT3–HIF-2α) link elevated iron status to hypoxia/iron-responsive transcription and cancer immune evasion [20,110,111,112,113,114].

Iron metabolism also directly affects tryptophan catabolism, which is a major immunosuppressive pathway in CRC. Iron-regulated STAT3 and NF-κB turn on IDO1, which breaks down tryptophan into kynurenine. Higher levels of kynurenine inhibit the growth of T cells and promote the growth of Tregs. Hence, the inflammation–iron–immune axis functions as a three-part regulatory module that promotes tumour survival and facilitates immune tolerance within the tumour microenvironment. Xue et al. [110] revealed that disrupting this axis—by limiting iron availability, blocking STAT3 signalling, or inhibiting IDO activity—may improve immune surveillance within the iron-enriched microenvironment of CA-CRC. Although evidence from cell culture and animal models strongly implicates iron overload and chronic inflammation in the pathogenesis of CRC, clinical and epidemiological studies in human patients have produced inconsistent results. These findings have not established a consistent or definitive association between dietary iron intake, serum ferritin levels, and the risk of CRC. While systemic iron status is certainly important, other factors such as gut microbiota composition, host genetic background, and tissue-specific iron distribution may play an even more critical role in influencing colorectal cancer risk and progression [115,116]. Another challenge is that ferritin and similar commonly studied biomarkers may lack diagnostic specificity, as their levels can also be elevated in a range of inflammatory or liver-related conditions, not just CRC. Iron chelation may cause anaemia and immune system suppression, and inhibiting COX-2 for a longer duration raises the risk of cardiovascular disease [117]. Before introducing iron-targeted or anti-inflammatory therapy, it is necessary to validate predictive biomarkers and identify acceptable patient groups [118]. Furthermore, ferroptotic cancer cells may alter and influence immunological responses that target tumours, as well as mechanisms that govern immune evasion via iron. New experimental evidence suggests that autophagy is essential for the release of High Mobility Group Box 1 (HMGB1) by cancer cells undergoing ferroptosis. When cancer cells die, they release high mobility group box 1 (HMGB1) into the TME, which activates the innate immune system by interacting with several pattern recognition receptors (PRRs), such as Toll-like receptors 2 and 4 (TLR2, TLR4) and the receptor for advanced glycation end products (RAGE). Specifically, it has been demonstrated that HMGB1 release from dying cancer cells enhances antigen processing and presentation on dendritic cells (DCs) through the TLR4-MyD88 axis, a process that occurs during chemotherapy or radiotherapy [119,120].

The soluble ER-associated chaperone calreticulin (CRT) can be more easily translocated to the surface of tumour cells through ROS-mediated ferroptosis. The plasma membrane of stressed or dying cells exposes CRT, which acts as a powerful “eat-me” signal. To trigger robust anti-tumour immune responses, the phagocytic “eat me” signal calreticulin (CRT) promotes the engulfment of tumour-associated antigens by antigen-presenting cells. Finally, mounting data suggest that tumour cells provide the manufacture of eicosanoids—which have been shown to enhance antitumour immunity—with arachidonic acid (AA) during ferroptosis. On the other hand, prostaglandin E2 (PGE2) release may accompany tumour cell ferroptosis activation, allowing tumours to evade immune monitoring. Consequently, ferroptotic cell activation of a strong immunological response may be intrinsically hindered by PGE2 synthesis [121,122] (Table 4).

## 6. Gut Microbiota–CRC Crosstalk: Mutual Outcomes

The gut microbiota exerts profound effects on colorectal carcinogenesis, and the interaction is bidirectional.

### 6.1. Effect of CRC on the Gut Microbiota

CRC reshapes the intestinal microbial ecosystem. Tumour-associated hypoxia, altered mucus secretion, and persistent inflammation create a selective niche that favours pathogenic or opportunistic organisms such as *Fusobacterium nucleatum*, enterotoxigenic *Bacteroides fragilis*, and colibactin-producing *Escherichia coli*, while reducing the abundance of protective commensals, including butyrate-producing *Clostridia* and *Firmicutes* [131,132]. These changes reduce microbial diversity, compromise mucosal barrier integrity, and reinforce pro-inflammatory signalling in the tumour microenvironment.

### 6.2. Effect of the Gut Microbiota on CRC

Gut microbes actively influence CRC initiation and progression. *F. nucleatum* promotes tumour adhesion and immune evasion through β-catenin signalling [133], while ETBF secretes BFT toxin that induces DNA damage and inflammation [134]. Colibactin-producing *E. coli* strains cause double-strand DNA breaks and genomic instability [135]. In contrast, beneficial commensals such as *Lactobacillus* and *Bifidobacterium* produce short-chain fatty acids (SCFAs), especially butyrate, which strengthen epithelial integrity, regulate immune tolerance, and exhibit anti-neoplastic properties [136].

### 6.3. Microbiota–Iron–CRC Axis

Iron metabolism serves as a critical interface linking microbiota and CRC. Tumour niches with high iron availability favour the expansion of siderophore-producing bacteria, which outcompete host cells for iron and exacerbate oxidative stress [137]. Dietary iron intake, tumour-driven metabolic reprogramming, and antibiotic use further destabilise the microbial balance, creating a vicious cycle that promotes tumour growth and immune suppression.

### 6.4. Implications

Thus, the gut microbiota–CRC relationship is reciprocal: CRC modifies microbial diversity and composition, while microbial metabolites, toxins, and iron-scavenging strategies regulate tumour growth, genomic instability, and immune evasion. Targeting this bidirectional crosstalk—through microbiome-based biomarkers, probiotics, dietary modulation, or iron-targeted interventions—may offer novel diagnostic and therapeutic strategies.

## 7. Therapeutic and Diagnostic Implications

CRC is a complex disease, and research into the interplay between iron metabolism and inflammatory signalling may offer valuable insights into its underlying biology, as well as improve strategies for diagnosis and treatment. Inflammatory pathways, such as those involving IL-6, COX-2, and STAT3, interact with key iron regulators, including hepcidin, ferroportin, and ferritin. Dysregulation of these regulators in CRC offers opportunities for biomarker-guided precision interventions and therapeutic targeting. Gonzalez Acera et al. [138] have studied extensively on this. In our work, we examined one approach that utilises a systems biology method to identify promising targets, supported by transcriptome data from mouse models and human CRC tissues associated with inflammatory bowel disease. The diagnostic and therapeutic implications of these findings are discussed with particular attention to clinical feasibility and target validation. Previous studies suggest that strategies such as iron chelation, induction of ferroptosis, and modulation of inflammatory pathways may hold promise for CRC treatment [139]. Nevertheless, most of these approaches are still in the early stages of development, and further mechanistic studies and clinical validation are required to translate these observations into effective therapies.

There is limited clinical evidence supporting the effectiveness of iron-targeted therapies in colorectal cancer. Currently, evidence is largely limited to small clinical trials and observational studies, with no large randomised controlled trials demonstrating the efficacy or safety of this approach in CRC patients [139]. Many of the pathways discussed, such as the IL-6/STAT3-hepcidin crosstalk, are well-studied in vitro; however, there is a lack of direct evidence from human tumour samples of different CRC subtypes. These gaps highlight the need for robust translational studies and biomarker-guided clinical trials to validate these approaches before they can be recommended for routine clinical use [140].

### 7.1. Iron Chelation Therapies: Benefits and Challenges

Preclinical studies suggest that iron chelation therapy could decrease tumour growth and spread by inhibiting iron-dependent activities. For example, deferoxamine has been shown in vitro to arrest the cell cycle and induce apoptosis in CRC cell lines. These findings indicate a potential therapeutic avenue, though clinical validation remains limited [141,142,143]. Schwartz et al. [144] reported that iron chelators may be particularly effective in tumours with high hepcidin and low ferroportin levels, as these tumours sequester iron intracellularly, rendering them more susceptible to iron depletion.

### 7.2. Anti-Inflammatory Therapies (Targeting IL-6, COX-2, etc.)

Inflammatory pathways, including IL-6/STAT3 and COX-2/PGE2, sustain chronic inflammation in the colonic mucosa of CRC patients. These pathways not only alter iron-regulating genes such as HAMP and FPN but also promote epithelial proliferation and apoptosis resistance. IL-6, via STAT3, increases HAMP expression, resulting in elevated intracellular iron and decreased ferroportin levels [145]. Inhibiting IL-6 may reduce HAMP expression and iron retention in human systems; however, human anti-IL-6R antibodies, such as tocilizumab, do not block IL-6 signalling in standard murine models. Preclinical evaluation, therefore, requires humanised IL-6R mice or murine-active IL-6/IL-6R inhibitors [146]. COX-2 inhibitors, such as celecoxib, reduce the expression of pro-inflammatory mediators, including IL-6, TNF, and prostaglandins. COX-2 activity also influences local hypoxia and HIF-1α activation, which in turn modulates iron transporter gene expression. Network analyses have identified ACO1, TFRC, and FTH1 as iron regulatory elements interacting with COX-2–related genes, suggesting that COX-2 inhibition may normalise both inflammatory and iron-overload–related gene expression profiles. Emerging evidence indicates that proteins such as ferritin and hepcidin may serve as diagnostic or prognostic biomarkers; however, their specificity is limited due to elevation in other inflammatory or hepatic conditions. Larger clinical studies are required before they can be adopted as reliable CRC biomarkers [147].

### 7.3. Diagnostic Biomarkers: Ferritin, Hepcidin, Gene Signatures

Iron-related proteins represent promising biomarkers for CRC risk assessment and treatment monitoring. Serum ferritin, for example, reflects both systemic iron stores and acute-phase responses, rising in response to pro-inflammatory cytokines such as IL-6 and TNF-α [148]. In patients with IBD and colitis-associated CRC (CA-CRC), elevated ferritin levels have been associated with poorer prognosis, higher tumour burden, and chemotherapy resistance. Hepcidin, measurable in blood and stool, is another potential biomarker of iron-overload–associated cancer, with high mucosal HAMP expression correlating with more aggressive histological features and reduced survival [149].

At the transcriptomic level, a curated inflammatory gene signature, including HAMP, IL6, FTH1, SLC11A2, and TNFAIP3, is more frequently observed in dysplastic tissue than in inflamed but non-dysplastic mucosa. This signature is much more common in dysplastic tissue samples than in inflamed but non-dysplastic mucosa. This molecular signature could help identify cancerous changes in long-term IBD early on and may also be used to predict the effectiveness of iron-targeted therapy [150]. Additionally, combining iron- and inflammation-related gene signatures with markers of epithelial proliferation (e.g., MKI67) and DNA damage (e.g., γ-H2AX) could form a highly specific multi-analyte panel for detecting CRC in high-risk populations [143].

### 7.4. Integration into Precision Medicine Approaches

Emerging evidence highlights the dysregulation of iron metabolism in CRC and its interplay with immune pathways, suggesting potential avenues for patient stratification and targeted therapy. Co-expression analysis of tissue samples from patients showed that there were different molecular endotypes, each with high levels of immune exhaustion markers like PDL1 and IDO1, as well as FTH1, HAMP, and IL6 [151].

Targeting HAMP in combination with checkpoint inhibition (using anti-PD-1/PD-L1) could prevent iron from suppressing the immune system and restore T cell cytotoxicity. Also, ferroptosis inducers like erastin or RSL3 could be more harmful to tumours that have a lot of ferritin and not much GPX4 activity [152]. These findings suggest that iron-rich, immunosuppressive tumour phenotypes could potentially respond to combined strategies (e.g., iron chelation plus immunotherapy). However, these approaches remain speculative, and rigorous clinical trials are needed to determine their safety and efficacy.

Stratifying patients in clinical trials based on iron metabolism–related biomarkers—including high HAMP/low FPN, elevated FTH1, or IL-6–dominant inflammatory profiles—could enable more personalised and effective therapeutic strategies. Such trials could facilitate personalised treatment regimens that integrate iron depletion, anti-inflammatory therapy, and immune checkpoint blockade. By aligning therapeutic strategies with the tumour’s metabolic-inflammatory landscape, this precision medicine approach goes beyond traditional TNM staging and may improve patient outcomes, IL6 [151,152].

## 8. Limitations and Research Gaps

A major limitation of the current evidence is the lack of correlation between preclinical findings and clinical outcomes, which hinders the translation of experimental insights into effective patient therapies. Translation from cell culture and animal models to patient treatment is still a challenge, even though most fundamental insights into inflammation and iron metabolism in CRC have been gained from these sources. Several factors hinder the translation of laboratory findings into clinical settings, including differences in the tumour microenvironment, genetic heterogeneity, and variability in host inflammatory responses. Additionally, while biomarkers such as ferritin and hepcidin show promise in experimental settings, their diagnostic accuracy and prognostic value in clinical practice remain uncertain and require further validation. These uncertainties highlight the risks of premature clinical extrapolation and the importance of well-designed translational research and randomised clinical trials before iron-targeted or anti-inflammatory treatments may be integrated into standard CRC therapy.

### 8.1. Limitations of Current Studies and Evidence

Although limitations remain in the current body of evidence, this review offers valuable insights into the underlying mechanisms of CRC, particularly the roles of iron metabolism, inflammation, and immune regulation. To begin, the intricacy of human CRC may be beyond the scope of preclinical models such as cell lines and mouse studies that establish connections between inflammation, iron metabolism, and CRC. There is a lack of large, long-term studies on how dietary iron intake, systemic iron overload, and inflammatory profiles work together to affect the risk of CRC [153]. There is also conflicting evidence about the role of systemic iron status. For example, some epidemiological studies have found no clear link between serum ferritin levels and the risk of CRC [153,154]. These disparities highlight the importance of conducting meta-analyses of prospective cohorts and randomised controlled trials to clarify causal relationships. A major challenge remains the lack of standardised biomarkers to assess iron dysregulation in CRC patients. Although ferritin, transferrin saturation, and hepcidin are commonly measured, their specificity for reflecting tumour-associated iron metabolism remains uncertain. Future studies should focus on combining multi-omic data with clinical outcomes to find robust biomarkers that can help with patient stratification and therapy selection [154].

### 8.2. Future Directions

CRC is a type of cancer that results from a combination of genetic predisposition, environmental exposures, metabolic alterations, and immune system dysregulation. From DNA damage and epithelial proliferation to immune evasion and resistance to cell death, this review reveals that iron metabolism and inflammatory signalling do not work independently but form a co-evolving and self-reinforcing axis that controls every significant sign of CRC progression. One of the key insights that emerges from transcriptomic, epigenomic, and proteomic studies is the presence of a deregulated iron-inflammation circuit. This circuit involves higher levels of HAMP, FTH1, TFRC, and SLC11A2, as well as increased activity of IL-6/STAT3, TNF/NF-κB, and COX-2/PGE2. These gene networks not only alter iron trafficking and storage in epithelial and immune cells but also significantly reshape the tumour immune microenvironment, leading to T cell exhaustion, increased resistance to ferroptosis, and impaired antigen presentation. This negative feedback loop is controlled by cytokines such as IL-6, IL-1β, and TNF-α, which in turn regulate nodal genes including HAMP (hepcidin), SLC40A1 (ferroportin), TFRC, and FTH1. Maintaining stable iron levels and suppressing immune system activity is a multi-system process. Histone modifications regulated by iron-dependent JmjC demethylases, promoter hypermethylation, and microRNA regulation targeting hepcidin (miR-122 and miR-485-3p) are all examples of such processes. At the same time, miR-21 and miR-155 boost STAT3 and NF-κB signalling while making intracellular iron stores less stable. This connects post-transcriptional control directly to tumour growth and immune escape. These gene networks work together to define a CRC subtype that has metabolic plasticity, chronic oxidative stress, and immunological coldness. These are all signs that cancer does not respond well to standard chemotherapy and immunotherapy.

Despite the abundance of molecular data, a major challenge remains in translating this knowledge into effective diagnostic and therapeutic strategies. The field requires integrated multi-omic models that combine somatic mutation data with epigenetic modifications, miRNA profiles, and iron regulatory signatures to more effectively track disease progression and evaluate treatment responses. Gene panels such as the HAMP–FTH1–SLC11A2 axis show potential as biomarkers for high-risk, inflammation-driven CRC; however, they require large-scale validation in prospective longitudinal cohorts before clinical implementation. Also, using spatial transcriptomics and iron-sensitive imaging techniques (like T2*-weighted MRI) to study the spatial and temporal dynamics of iron distribution in tumour, stromal, and immune compartments could completely change how we think about iron zoning in the tumour microenvironment.

Future treatment plans should target both iron metabolism and inflammation simultaneously. Iron chelators, such as deferoxamine or Dp44mT, when used in conjunction with JAK/STAT or COX-2 inhibitors, may help reduce the two types of cancer-causing pressures caused by excessive iron and chronic inflammation. Additionally, tumours with a high expression of FTH1 and IDO1 may be particularly susceptible to combination strategies involving ferroptosis inducers and immune checkpoint inhibitors, offering a potential avenue for targeted therapy. Inhibitors of DNA methyltransferase and anti-miR-21 are examples of epigenetic treatments that may make tumours more susceptible to immunological cytotoxicity and ferroptosis. The iron–inflammation axis is an immunologically strategic network that is transcriptionally dynamic, epigenetically stable, and essential to several pathobiological processes in CRC, especially inflammation-driven subtypes. Advanced precision medicine in CRC requires the development of therapeutics that simultaneously target multiple pathways, including genetic, immunological, and metabolic mechanisms. Uncovering the molecular basis of this axis could fundamentally transform CRC treatment. This approach may enable the creation of biomarker-driven early detection systems and more effective, personalised therapies.

## Figures and Tables

**Figure 1 genes-16-01166-f001:**
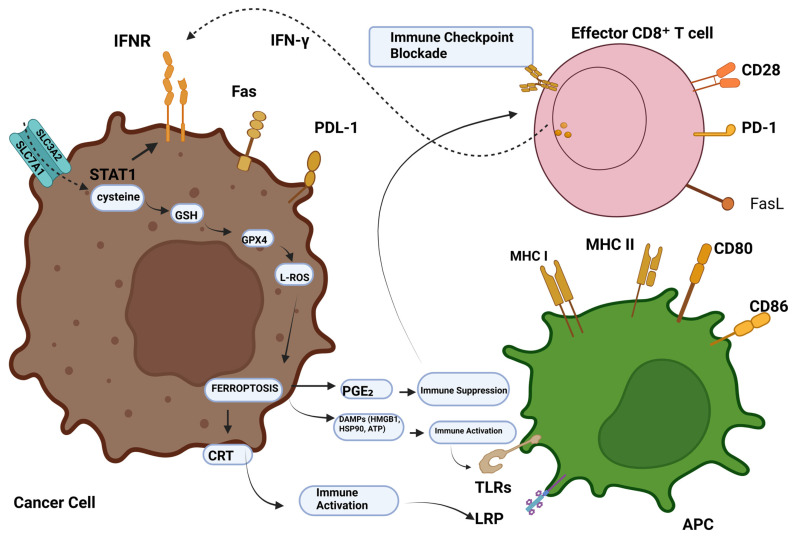
Ferroptosis-tumour immunity modulation. CD8^+^ T cells stimulated by immunotherapy emit IFN-γ, activating INFR. INFR suppresses STAT1-mediated SLC7A11 transcription. This downregulation of system Xc- causes ROS-mediated ferroptosis via GSH-GPX4. Ferroptotic cells generate DAMPs like HMGB1, CRT, and AA, which recruit and activate immune cells. Conversely, elevated PGE2 helps tumour immune evasion. Abbreviations: IFN-γ, INFR, SLC7A11, SLC3A2, STAT1, ROS, GSH, GPX4, DAMPs, HMGB1, calreticulin, AA, PGE2, and PGE2 [14]. Reproduced via Biorender from Ref. [14] with permission under a Creative Commons Attribution License (CC-BY), copyright the authors.

**Figure 2 genes-16-01166-f002:**
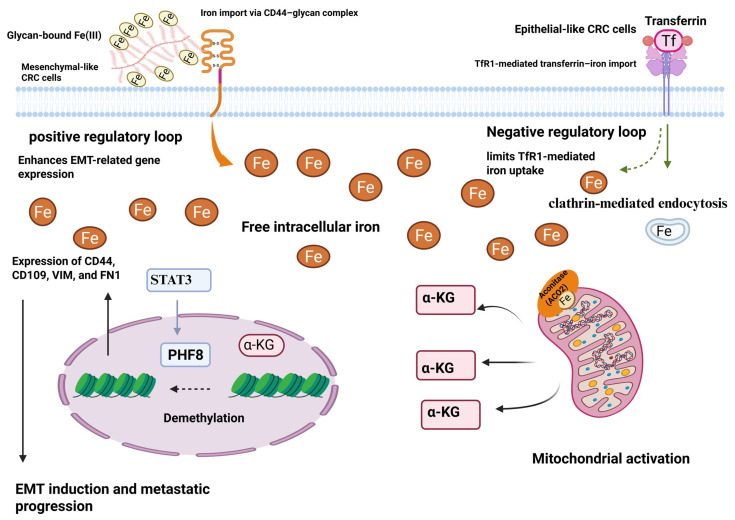
CD44 plays a role in the uptake of iron during EMT in colorectal cancer cells. TF-TfR1 is the mechanism by which epithelial phenotype colorectal cancer cells import iron. Intracellular iron and TfR1 form a negative feedback loop that limits the TF-TfR1 pathway, meaning that mesenchymal phenotype CRC cells require more iron. The absorption of iron by CRC cells through a glycan-Fe (III)-CD44-mediated process can enhance mitochondrial function and the generation of α-ketoglutaric acid (α-KG). The activity of the plant homeodomain finger protein 8 demethylase is enhanced by an increase in α-KG in conjunction with ferrous iron. In the end, this leads to the expression of genes related to EMT, such as CD44 [20]. Reproduced via Biorender from Ref. [20] with permission under a Creative Commons Attribution License (CC-BY), copyright the authors.

**Figure 3 genes-16-01166-f003:**
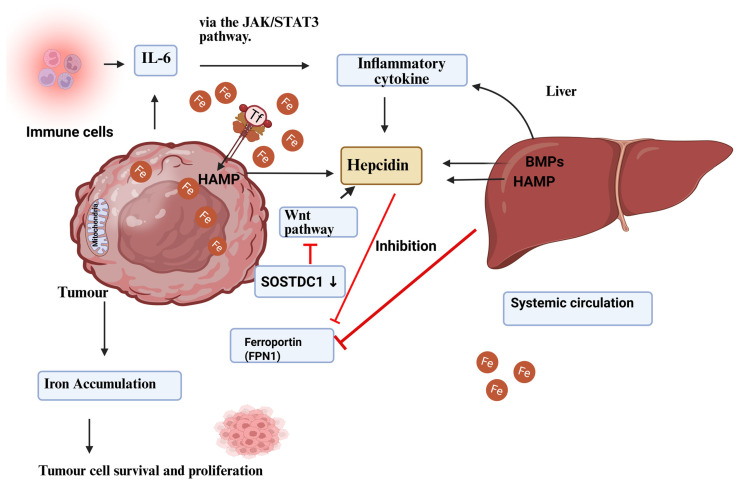
Regulation of cancer hepcidin production. Hepcidin is produced by the liver and malignant tumour cells. Hepcidin overexpression in tumour tissue is linked to BMP molecules, such as BMP7, and inflammatory triggers, including interleukin-6. In cancer tissue, the Wnt pathway and the sclerostin domain-containing protein 1 (SOSTDC1), which is downregulated due to epigenetic silencing, have been found to regulate hepcidin. Ferroportin sequesters iron in tumour cells when hepcidin levels rise in the tumour microenvironment. Hepcidin increases with TFR1 overexpression, which enhances tumour cell iron availability. Increasing iron stores helps tumour cells survive and grow [97]. Reproduced via Biorender from Ref. [97] with permission under a Creative Commons Attribution License (CC-BY), copyright the authors.

**Figure 4 genes-16-01166-f004:**
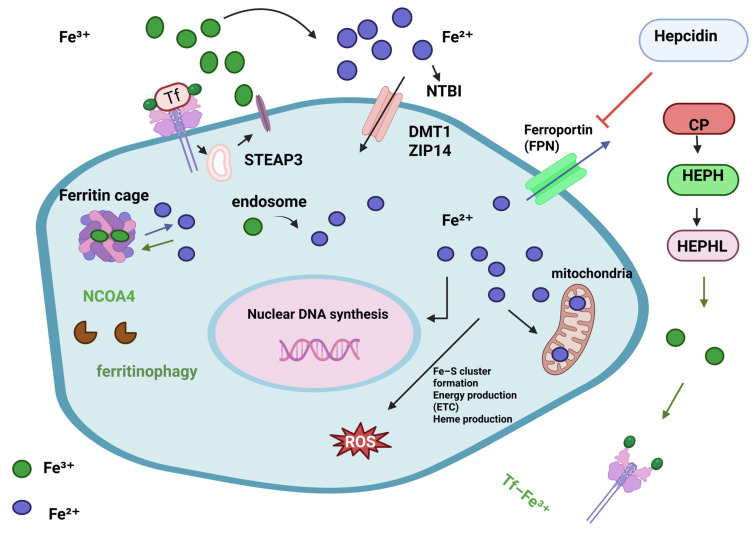
Cellular iron metabolism. Iron levels are closely managed since too little or too much can harm cells. Iron intake, use, storage, and export must be synchronised, as must the cell’s conversion between Fe^2+^ and Fe^3+^ oxidation states. Iron importers DMT1 and ZIP14 take up TF-Fe^3+^ and NTBI into the cell. STEAP3, a ferrireductase, converts Fe^3+^ to Fe^2+^ for import. Once inside the cell, bioavailable and more soluble Fe^2+^ is used for DNA replication, ROS production via Fenton/Haber-Weiss (F/H-W) chemistry, mitochondrial bioenergetics, Fe-S and heme biosynthesis, and by many proteins. Due to ROS generation, excess Fe^2+^ iron is harmful. Thus, it must be stored but accessible. Ferritin proteins create the “ferritin cage” to store the insoluble Fe^3+^ form of iron. NCOA4 destroys this ferritin cage to release iron when intracellular levels are low. aturated intracellular iron levels require iron export. Ferroportin (FPN) exports iron. Fe^2+^ iron is oxidised to Fe^3+^ outside the cell by CP, HEPH, HEPHL. After binding to transferrin (Tf-Fe^3+^), Fe^3+^ iron enters circulation to restart the cycle. Hepcidin, a liver hormone, regulates iron metabolism. Hepcidin degrades FPN in cells to prevent iron release into the blood when systemic iron levels are high. Alternatively, low iron blood levels limit hepcidin expression [108]. Reproduced via Biorender from Ref. [108] with permission under a Creative Commons Attribution License (CC-BY), copyright the authors.

**Table 1 genes-16-01166-t001:** Key Regulators of Iron Metabolism in CRC and Their Functions.

Regulator	Gene Symbol	Cellular Location	Normal Function	Regulation Mechanism	Alteration in CRC	Impact on Tumour Biology	References
Transferrin Receptor 1	TFRC (TfR1)	Plasma membrane	Imports Fe^3+^ bound to transferrin via endocytosis	IRP1/2 stabilises TFRC mRNA under low iron	Upregulated	Increases iron uptake, promoting DNA synthesis and proliferation	[45]
Ferroportin	SLC40A1 (FPN)	Plasma membrane	Exports Fe^2+^ from cells into circulation	Downregulated by hepcidin binding and degradation	Downregulated	Reduced iron efflux, increasing intracellular iron and oxidative stress	[46,47]
Ferritin (H- and L-chains)	FTH1/FTL	Cytoplasm (also secreted)	Stores Fe^3+^ in a non-toxic form	Translation suppressed by IRP1/2 under low iron	Altered levels (often decreased intracellularly)	Reduced storage increases free iron pool and ROS production	[48,49]
Iron Regulatory Proteins	IRP1/IRP2 (ACO1/IREB2)	Cytoplasm	Bind to iron-responsive elements (IREs) on mRNA to regulate TFRC, FPN, and ferritin	Sensing iron–sulphur cluster or iron availability	IRP2 is often upregulated	Stabilises TFRC mRNA, represses ferritin and FPN, boosting iron availability	[50,51]
Hypoxia-Inducible Factor-1α	HIF1A	Nucleus (active under hypoxia)	Transcription factor regulating hypoxic responses and iron metabolism genes	Stabilised under hypoxia; degraded in normoxia. HIF-1α regulates transcription, while HIF-2α mRNA contains an IRE and is regulated post-transcriptionally by IRPs.	Upregulated in hypoxic CRC regions	Promotes TFRC and DMT1 expression, suppresses FPN/ferritin, increasing labile iron poo	[52,53]

**Table 2 genes-16-01166-t002:** Major and Emerging Inflammatory Pathways Driving CRC Progression.

Pathway/Factor	Upstream Stimuli	Activation Mechanism	Key Downstream Effects	Pathway Crosstalk	Impact on CRC Progression
IL-6 → STAT3	Chronic inflammation, CAFs, microbiota	IL-6 → IL-6R/gp130 → JAK2-STAT3 phosphorylation	Induces Cyclin D1, c-Myc, BCL-XL, survivin, MMPs, VEGF	Feedback with NF-κB, IL-23, Wnt/β-catenin	Drives proliferation, survival, EMT, metastasis [71]
TNF-α → NF-κB	Immune cells, tumour inflammation	TNF-α binds TNFR1/2 → IKK activation → IκB degradation → NF-κB nuclear translocation	Upregulates Bcl-2, cIAPs, IL-6, chemokines, MDR1	Engages STAT3, IL-6, COX-2	Promotes survival, inflammation, and chemoresistance [72]
NF-κB	TNF, IL-1β, TLRs, oxidative stress	IκB degradation → NF-κB activation (p65/p50 or RelB/p52)	Activates cytokines, angiogenesis, EMT, and chemoresistance genes	Central node linking TNF, IL-1β, TLR, COX-2-2 and IL-6	Sustains inflammation, tumour progression, therapy resistance [73]
COX-2 → PGE_2_	NF-κB, IL-1β, TNF, TLR4	COX-2 enzyme converts arachidonic acid to PGE_2_ → EP receptor signalling	PI3K/AKT activation, VEGF, and immunosuppressive cell recruitment	Upstream NF-κB & IL-1β; enhances IL-6/STAT3 axis	Fosters tumour growth, angiogenesis, immune evasion; NSAIDs protective [74]
IL-1β → NF-κB/AP-1	Inflammasome-matured macrophages	IL-1β binds IL-1R → MyD88-dependent NF-κB and AP-1 activation	Induces COX-2, IL-6, IL-8, EMT, proliferation	Amplifies NF-κB, COX-2, STAT3	Enhances inflammation-driven tumour initiation and chemoresistance [75]
TLR4 (MyD88/TRIF)	Microbial LPS, DAMPs	TLR4 ➝ MyD88 (pro-inflammatory)/TRIF (IFNs) → NF-κB, IRF3	Releases IL-6, TNF, CCL2/CCL20; recruits TAMs/MDSCs	Activates NF-κB, STAT3, Wnt pathways	Promotes immune infiltration, metastasis, and poor outcomes [76]
IL-23 → STAT3 → IL-17	TAMs activated by TLRs, microbial signals	IL-23 binds IL-23R → JAK2/TYK2 → STAT3 → promotes Th17 cell activation and IL-17 production	Drives IL-17A/F secretion, MMPs, VEGF, STAT3 in tumour/stromal cells	Cooperative with IL-6-6/STAT3 and TLR4	Correlates with rapid CRC progression, metastasis, and poor prognosis [77]
IL-17A/F (Th17, γδ T cells, ILCs)	Driven by IL-23, IL-1β, and microbiota	IL-17 binds IL-17R → Act1/TRAF6 → NF-κB, MAPK, STAT3	Promotes IL-6, VEGF, CXCL1/8, recruitment of MDSCs/Tregs and suppresses CD8^+^ T cells	Enhances STAT3, NF-κB, IL-6 loops	Promotes angiogenesis, immunosuppression, tumour growth and metastasis [78]
IL-33 (alarming)	Released by necrotic epithelial or stromal cells	IL-33 binds ST2 receptor → NF-κB/MAPK activation	Induces Th2 cytokines (IL-4/5/13), recruits mast cells and ILC2s	Signals via NF-κB, interacts with TLRs and STAT pathways	May modulate tumour microenvironment, Treg recruitment and immune modulation in CRC [79]
TLR5 → NF-κB/TNF-α	Recognition of bacterial flagellin	TLR5 binds flagellin → MyD88/IRAK/TRAF6 → NF-κB → TNF-α, IL-8	Induces pro-inflammatory cytokines CXCL8, TNF-α; recruits inflammatory cells	Activates NF-κB, TNF pathways; links microbiota sensing to cytokine cascade	Enhance inflammation, immune cell recruitment and CRC initiation [80]

**Table 4 genes-16-01166-t004:** Iron–Inflammation Crosstalk in CRC.

Component/Axis	Upstream Trigger(s)	Molecular Mechanism	Cellular Outcomes	Tumour & Immune Effects in CRC
IL-6 → STAT3 → Hepcidin	Chronic inflammation, gut dysbiosis	IL-6 → JAK/STAT3 → hepcidin up → ferroportin down	Iron retention in TAMs & epithelial cells; systemic anaemia	Local iron-rich niche; ROS generation, DNA damage, NF-κB induction; self-sustaining IL-6/hepcidin inflammation loop [123]
IL-1β/TNF-α → DMT1, NRAMP1	TAMs, stromal inflammation	Upregulate iron importers DMT1 and SLC11A1 in TAMs/epithelial cells	Enhanced iron uptake and trapping in tissue	Supports STAT3 phosphorylation, anti-apoptotic phenotype, tumour survival under chronic inflammation [110]
Ferritin (FTH1/FTL)	Elevated intracellular iron plus inflammatory signals	Intracellular storage; secreted ferritin binds Tim-2/Scara5 on immune cells	Impairs DC maturation; polarises Tregs/Th17; suppresses CD8^+^ and NK cytotoxicity	Constitutive immune suppression; antigen-presentation blockade in IBD-associated CRC [124]
TFRC (Transferrin Receptor-1)	IRP2 activation with inflammation; hypoxia via HIF-1α	Enhanced TFRC surface expression boosts iron import	Enriched labile iron pool in tumour cells	Promote proliferation, metabolic flexibility, genomic instability via ROS [125]
Heme Iron → Gut Epithelium Dysregulation	High dietary red meat, heme iron metabolism	Heme iron damages the epithelial barrier; promotes ROS & inflammatory cytokines	Epithelial injury, increased permeability, inflammation	Drives CRC risk; contributes to dysbiosis and inflammation-linked tumourigenesis [126]
CD44-mediated iron endocytosis	EMT, TGF-β signalling, inflammation	Upregulated CD44 imports iron bound to hyaluronan complexes	Iron influx independent of TFRC; epigenetic remodelling	Enhances EMT, metastasis potential, and metabolic plasticity in CRC cells [127]
Microbiota dysbiosis + TLR signalling	Iron-altered microbiome; LPS, DAMPs	TLR4/TLR5 activation → NF-κB, IL-6/IL-1β release	Mucosal inflammation, cytokine secretion, and macrophage recruitment	Iron-modulated microbiome triggers inflammatory cytokines and CRC-promoting microenvironment [128]
ROS → NF-κB/STAT3 loops	Increased LIP from TFRC/hepcidin; inflammation	Fenton chemistry yields hydroxyl radicals; triggers NF-κB and STAT3 signalling	DNA damage, sustained cytokine production, and genomic instability	Supports chronic inflammation, proliferative signalling, tumour progression [129]
Immune checkpoint induction (IDO1, PD-L1)	STAT3 & NF-κB activation in iron-rich contexts	IDO1/TDO and PD-L1 gene upregulated by STAT3, NF-κB	Kynurenine accumulation; T-cell exhaustion; increased Tregs	Promotes immune evasion and tolerance in iron-rich tumour microenvironment [130]

## Data Availability

No additional data available.

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
