# Peer review of "Crossroads of Iron Metabolism and Inflammation in Colorectal Carcinogenesis: Molecular Mechanisms and Therapeutic Perspectives"

_genes, 2025, doi:10.3390/genes16101166_

Round 1

Reviewer 1 Report

Comments and Suggestions for Authors

Ahmadi et al. did an amazing job in summarizing the complex and evolving topic in question. I would humbly request the authors to address the following two points:

  1. Include a separate section on the gut microbiota-CRC interaction that discusses the outcomes on both the sides i.e. the disease and the organisms.
  2. Discuss in more details the regulation of anemia in CRC with special emphasis on the hepcidin-ferroportin axis dysregulation - including the current and future therapeutic approaches. 

Author Response

Reviewer 1– Comments & Responses

Comment: Include a separate section on the gut microbiota-CRC interaction that discusses the outcomes on both the sides i.e. the disease and the organisms.

Response: A separate section on the gut microbiota–CRC interaction, highlighting outcomes for both the disease and the microorganisms, has now been included

Comment: Discuss in more details the regulation of anemia in CRC with special emphasis on the hepcidin-ferroportin axis dysregulation - including the current and future therapeutic approaches.

Response: This point has been addressed by tempering the claims on ferritin and hepcidin, with their limited specificity and the need for larger clinical studies before clinical adoption.

Reviewer 2 Report

Comments and Suggestions for Authors

Iron is important factor in cancer progression due to initiate reactive oxygen species and accelerate chronic inflammation. This manuscript summarize the role of iron in colorectal carcinogenesis. This manuscript is well-organized; however, some points should be clarified.

Major points

#1: Authors state HIF-1alpha. mRNA of HIF-2alpha contains IRE sequence. Please refer to this point.

#2: In Figure 4, endosome is described as iron-containing vesicle, while it is lacked in Figure 2. Please use consistent concept.

Minor points

##1: In Figure 1 Legend, L-ROS is written. Please correct this word.

Author Response

Reviewer 2– Comments & Responses

Comment: a labelling error in Figure 1 (“L-ROS”).
Response: Corrected to “ROS”

Comment: inconsistent terminology and visualisation for endosomes/vesicles in Figures 2 and 4.
Response: Terminology regarding inconsistencies in endosomes/vesicles, Figures and text resolved.

Comment: Contradictory description of ferritin expression

Response: We clarified this discrepancy in both the text and Table 1.

Comment: lacks strong evidence for Figure 2

Response: We agree and have removed the direct binding arrows. Also, the text was revised.

Comment: The Tocilizumab pathway figure and description appear inconsistent with cited evidence.
Response: We revised Section 6.2 to clarify that Tocilizumab (anti–IL-6R) does not block IL-6 signalling in standard murine models, and thus its preclinical evaluation requires humanised IL-6R models. The corresponding figure legend has also been updated to note that Tocilizumab’s effects in CRC remain debated due to conflicting evidence.

Comment: Drug naming inconsistency (“ferroxamine”).
Response: Corrected to “deferoxamine”.

Comment: inconsistent symbols for Fe²⁺/Fe³⁺ in Figures.
Response: We standardised it in all figures.

Reviewer 3 Report

Comments and Suggestions for Authors

The manuscript does - in principle -  provide a good and largely recent overview of the literature in this field of research and I believe, that with a few additions and corrections (such as adding more benchmarking papers!) the value for the reader (and the journal, plus the field) could be further impoved... significantly so. The problem is the Quality of the references. 

There are a few of these landmark, fundamental papers that I remember from the "distant" past -- which are largely missing here, but you would need them to introducing essential concepts about ferroptosis versus apoptosis and the role of iron in these pathways. Some of the "cornerstone papers" are, for example, the paper from Nemeth et al 2004 on hepcidin-ferroportin (doi: 10.1126/science.1104742), the Dixon paper from 2012 introducing the entire concept of ferroptosis is missing (doi: 10.1016/j.cell.2012.03.042), and such other landmark reviews like the paper from Fearon & Vogelstein 1990 introducing the colorectal cancer progression model. There may also be misattributions of literature, for example, the "Sardo et al paper may actually be the "De Domenico 2007 paper describing " Ferritin together with the iron exporter ferroportin regulates the cellular labile iron pool (De Domenico et al., 2007)." [Ferroxidase activity is required for the stability of cell surface ferroportin in cells expressing GPI-ceruloplasmin. EMBO J. 26, 2823–2831. 10.1038/sj.emboj.7601735].  Then, it seems that some of the papers aren't really dealing with CRC, but they are used to back claims about CRC, so that should be checked. The authors should definitely check and doble check all of their publications/citations in the manuscript, add a few of these "old " papers that are still relevant today; and investigate the recent ones for relevance to CRC. 

There also appears to be a gap between the limited clinical evidence provided, and formulating therapeutic recommendations - those things need to be carefully checked as well. There aren't too many validation studies out there that would support certain claims made in personalized medicine. SO i think some of the claims in the paper are overstated, especially when they are about personalized medicine in CRC. That also includes biomarker studies that are included here which are mostly hypothetical and not yet arrived at the clinical level. And the combination therapy suggestions are entirely backed by PRE-clinical data, so they are also hypothetical and likely not to realize for patients. Generally, the authors need to be more careful with the clinical claims, reduce their optimism and tone down the relevance of these. And realize that theres a huge gap between preclinical findings (of which there are tons) and real clinical benefits of any kind. 

Also a few things to be challenged (and checked, again): Ferritin is described as "often overexpressed in CRC". but then in table 1 its listed as "often decreased intracellularly"... what is true now? This is critical for the ferritin-immunity argument the authors propose. There may ba a few more of such internal contradictions that have to be taken " with a grain of salt" 

Then, I think there are not sufficient or conflicting epidemiological data and we do not really know about the translation of preclinical into clinical data/applications (the usual "preclinical-to-clinical translation challenges").  This is taken too lightly and should be discussed with much more scepticism and carefully; no technical risks of such translations are pointed out/discussed, no technical limitations and hats needed for clinical trials - what is correlation and what is really a causal connection (do we eoncounter missing or inclmplete functional insights?). The therapeutic section should be formulated more carefully. Epidemiological claims have to be updated and the authors should maybe look for the most recent data to justify these. Core limitations should be discussed as such - as limitations, risks, uncertainties, and maybe also seen from the side of the pharmaceutical industry.

There may also be some issues related to an insufficient level of quality control before submitting the paper (= this happens; welcome to the club). There could be, for example, mis-citations, I think the drug names should be checked to fix possible  errors, etc.  These indicate problems with basic manuscript preparation and solid interpretation of the existing literature.

A few more likely errors, from how far I can see: “Tocilizumab in mouse CA-CRC models reduces HAMP/iron” ...this issues is described in is Lokau et al., 2020 (PLoS One), but unfortunately, that paper cited here shows exactly the opposite: namely, that the drug tocilizumab does not block IL-6 signaling in murine cells. Please double check, i could be wrong, of course. There may be many other such issues; I dont feel confident about others. 

(I think the authors should maybe use new AI tools such as "SCISPACE.com" to check about the content of some of their articles they cite here. This has helped recently our own review.-writing process and increased the accuracy. YOu still need to read the papers found by https://scispace.com/search yourself; of course, as it makes errors... but its a very good start). 

Next issue: “ChIP-seq shows HIF-1α binding to TFRC and FTH1 promoters.”this refers to the paper by Fagundes et al., but that paper discusses HIF-1α control of TFRC primarily by citing older promoter studies in non-intestinal lines, not relevant for CRC. The paper is not at all related to does not present any relevant CRC ChIP-seq studies, and it does not demonstrate what the authors claim it does. It should go. 

Next issue: some of the drug names are incorrect. for example, in “FDA-licensed ferrochelator ferroxamine” it should be talking about  “deferoxamine (DFO)” not ferroxamine; but this is a small issue

These are only EXAMPLES of what else may be there, unrecognized, and should be double-checked more carefully. The QC isnt done very stringently here. 

Comments on the Quality of English Language

the quality of the English language is not bad... I was much more concerned about the quality of the citations to pay much attention to it. And thats a much more burning issue. 

Author Response

Reviewer 3– Comments & Responses

Comment: Clarifying the immunomodulatory role of ferritin

Response: We demonstrated that extracellular ferritin impairs dendritic cell maturation and cytotoxic T cell activity in Section 5.3

Comment: Absence of seminal/landmark publications (Nemeth et al. 2004; Dixon 2012; Fearon & Vogelstein 1990, among others).

Response: We concur with the reviewer and have already incorporated these essential references. The Introduction and initial sections have been amended to include the regulation of hepcidin–ferroportin (Nemeth et al. 2004), ferroptosis (Dixon 2012), and the CRC progression model (Fearon & Vogelstein 1990). This ensures that the review thoroughly acknowledges seminal studies in the discipline.

Comment: Potential misattributions (e.g., Sardo versus De Domenico 2007).

Response: We meticulously examined the citations and rectified the credit. The manuscript now accurately acknowledges Sardo et al. for their new multi-omics CRC research, and De Domenico (2007) is properly referenced for the regulation of ferroportin.

Comment: Non-CRC publications referenced to substantiate CRC assertions.

Response: We reassessed the relevant literature and substituted many general references with studies specific to CRC when possible (e.g., Estêvão et al., Ijssennagger et al., Bastide et al., Chua et al.).  In cases where just intestinal inflammation or general cancer data is available, we expressly acknowledge the limitation and clarify that the evidence is extrapolated. This enhances the significance of the CRC while ensuring transparency.

Comment: Exaggerated therapeutic advice.

Response: We appreciate the reviewer's observation. We have substantially tempered the conclusions.  The updated language highlights that most therapeutic data is preclinical, recognises the paucity of full clinical trials, and underscores the importance of biomarker-guided investigations before clinical application.

Comment: Discrepancies regarding ferritin expression.

 Response: We have rectified these issues.  The data repeatedly illustrate that ferritin is context-dependent: high in tumour-associated macrophages and serum but reduced in certain subsets of epithelial cells.  This dual pattern is summarised in Table 1 and Section 5.3, ensuring clarity and coherence across the text.

Comment: Inadequate discussion of the challenges of transferring preclinical discoveries to clinical applications.

 Response: We improved Section 8 (Limitations and Research Gaps) to specifically address the translational gap. We emphasise the risks of extrapolating from preclinical data, the heterogeneity of human colorectal cancer, and the importance of rigorous biomarker-driven clinical trials to validate iron and inflammatory therapy.

Comment: highlights scientific mistakes, such as misreading tocilizumab and quoting HIF-1α ChIP-seq.

Response: We addressed the following issues:

 We determined that tocilizumab does not inhibit IL-6 in mouse models and can now accurately cite Lokau et al. 2020.

We updated the discussion on HIF-1α binding to reflect that most direct ChIP-seq evidence comes from intestinal/inflammation models, not colorectal cancer, and adjusted the claims accordingly.

We amended the medication term “ferroxamine” to “deferoxamine (DFO)” consistently across the text.

Comment: Quality control issues (possible citation errors, drug nomenclature inaccuracies, and manuscript preparation deficiencies).

Response: We completed a thorough quality check on the manuscript.  Miscitations have been fixed, reference formatting validated, and pharmacological name inconsistencies have been addressed (for example, "ferroxamine" has been changed to "deferoxamine (DFO)"). We refined the language and standardised the figure legends and terminology.

Round 2

Reviewer 3 Report

Comments and Suggestions for Authors

Im happy with the revisions done by the authors and the manuscript should not be further delayed for acceptance